# The Need to Set up a Biobank Dedicated to Lymphoid Malignancies: Experience of a Single Center (Laboratory of Clinical and Experimental Pathology, University Côte d’Azur, Nice, France)

**DOI:** 10.3390/jpm13071076

**Published:** 2023-06-29

**Authors:** Christophe Bontoux, Aubiège Marcovich, Samantha Goffinet, Florian Pesce, Virginie Tanga, Doriane Bohly, Myriam Salah, Kevin Washetine, Zeineb Messaoudi, Jean-Marc Felix, Christelle Bonnetaud, Lihui Wang, Geetha Menon, Jean-Philippe Berthet, Charlotte Cohen, Jonathan Benzaquen, Charles-Hugo Marquette, Sandra Lassalle, Elodie Long-Mira, Veronique Hofman, Luc Xerri, Marius Ilié, Paul Hofman

**Affiliations:** 1Laboratory of Clinical and Experimental Pathology, Université Côte d’Azur, Hôpital Pasteur, CHU de Nice, CEDEX 1, 06001 Nice, France; aubiege.marcovich@hotmail.fr (A.M.); goffinet.s@chu-nice.fr (S.G.); tanga.v@chu-nice.fr (V.T.); bohly.d2@chu-nice.fr (D.B.); salah.m2@chu-nice.fr (M.S.); washetine.k@chu-nice.fr (K.W.); messaoudi.z@chu-nice.fr (Z.M.); felix.jm@chu-nice.fr (J.-M.F.); bonnetaud.c@chu-nice.fr (C.B.); lassalle.s@chu-nice.fr (S.L.); long-mira.e@chu-nice.fr (E.L.-M.); hofman.v@chu-nice.fr (V.H.); ilie.m@chu-nice.fr (M.I.); 2Hospital-Integrated Biobank (BB-0033-00025), Université Côte d’Azur, Hôpital Pasteur, CHU de Nice, CEDEX 1, 06001 Nice, France; 3Team 4, Institute of Research on Cancer and Aging of Nice (IRCAN), Inserm U1081, CNRS UMR7284, Université Côte d’Azur, CHU de Nice, CEDEX 2, 06107 Nice, France; 4FHU OncoAge, Université Côte d’Azur, CEDEX 1, 06001 Nice, France; benzaquen.j@chu-nice.fr (J.B.); marquette.c@chu-nice.fr (C.-H.M.); 5Institut Hospitalo-Universitaire (IHU), RespirERA, Université Côte d’Azur, Hôpital Pasteur, CHU de Nice, CEDEX 1, 06001 Nice, France; berthet.jp@chu-nice.fr (J.-P.B.); cohen.c@chu-nice.fr (C.C.); 6Department of Biopathology and Tumor Immunology, Institut Paoli-Calmettes, Centre de Recherche en Cancérologie de Marseille, INSERM U1068, Centre National de la Recherche Scientifique UMR 7258, Aix-Marseille University, UM105, CEDEX 9, 13273 Marseille, France; pescef@ipc.unicancer.fr (F.P.); xerril@ipc.unicancer.fr (L.X.); 7Haemato-Oncology Diagnostic Service, Cheshire & Merseyside Cancer Network, Liverpool University Hospitals NHS Foundation Trust, CSSB Building Level 4, Vernon Street, Liverpool L7 8YE, UK; lihui.wang@liverpoolft.nhs.uk (L.W.); geetha.menon@liverpoolft.nhs.uk (G.M.); 8Department of Thoracic Surgery, FHU OncoAge, Université Côte d’Azur, Hôpital Pasteur, CHU de Nice, CEDEX 1, 06001 Nice, France; 9Department of Pneumology, FHU OncoAge, Université Côte d’Azur, Hôpital Pasteur, CHU de Nice, CEDEX 1, 06001 Nice, France

**Keywords:** lymphoma, biobank, specimens, indicators, clinical data, translational research

## Abstract

Several therapies to improve the management of lymphoma are currently being investigated, necessitating the development of new biomarkers. However, this requires high-quality and clinically annotated biological material. Therefore, we established a lymphoma biobank including all available biological material (tissue specimens and matched biological resources) along with associated clinical data for lymphoma patients diagnosed, according to the WHO classification, between 2005 and 2022 in the Laboratory of Clinical and Experimental Pathology, Nice, France. We retrospectively included selected cases in a new collection at the Côte d’Azur Biobank, which contains 2150 samples from 363 cases (351 patients). The male/female ratio was 1.3, and the median age at diagnosis was 58 years. The most common lymphoma types were classical Hodgkin lymphoma, diffuse large B-cell lymphoma, and extra-nodal marginal zone lymphoma of MALT tissue. The main sites of lymphoma were the mediastinum, lymph node, Waldeyer’s ring, and lung. The Côte d’Azur Biobank is ISO 9001 and ISO 20387 certified and aims to provide high quality and diverse biological material to support translational research projects into lymphoma. The clinico-pathological data generated by this collection should aid the development of new biomarkers to enhance the survival of patients with lymphoid malignancies.

## 1. Introduction

Lymphoid malignancies, encompassing both Hodgkin and non-Hodgkin lymphomas, are relatively infrequent when compared to solid tumors. In the global context, they rank as the seventh most prevalent types of cancer, constituting approximately 3% of all cancer cases [1]. Specifically, in the year 2020, there were approximately 630,000 newly diagnosed cases, with a 5-year prevalence rate of 24 per 100,000 individuals [1]. It is worth noting that the overall survival (OS) rates vary significantly across different lymphoma histological subtypes, with Hodgkin lymphoma generally displaying a more favorable prognosis, while certain types of T cell lymphoma exhibit a considerably poorer outcome [2,3].

In the past few years, significant advancements have been made in the understanding of lymphomagenesis, accompanied by the development of innovative therapeutic approaches. These groundbreaking strategies include Chimeric Antigen Receptor (CAR) T-cell therapy, antibody-drug conjugates, bispecific T-cell redirectors, and agents targeting B-cell receptor signaling [4,5,6,7,8]. Despite these notable advancements, lymphoma continues to pose a serious health concern, as evidenced by a cumulative risk of death reaching 30% and an estimated 280,000 global deaths in 2020 [1,9]. Consequently, extensive research efforts are currently underway to investigate potential therapies and develop new prognostic and theranostic biomarkers [10,11]. Access to large numbers of clinically well-annotated biological resources (tissues and biofluids) of high quality collected from patients is a mandatory step in the discovery of novel mechanisms of tumorogenesis [12,13]. In addition, well-managed biological material is pivotal for the validation of predictive and prognostic biomarkers [14,15,16]. Nevertheless, to define reliable biomarkers, a relatively high number of patients samples must be collected and analyzed.

In this context, biobanking is a critical activity to collect enough high-quality samples with clinically associated data for translational research purposes. However, the heterogeneity of clinical practices makes the management and homogenization of the pre-analytical steps difficult to master [17]. The quality of the obtained samples varies greatly between structures and organizations and can lead to inconsistent and non-reproducible results that may affect the economic sustainability of the research [18]. Therefore, the need for high quality biological resources associated with clinical, histological, and molecular data is constantly increasing [19]. Consequently, biobanks must adapt their offer to satisfy the challenging requests of different stakeholders and the scientific community. Their management has to evolve over time to fulfill the objectives of research projects that constantly progress [20]. However, different constraints have to be met when using human samples for research purposes, including ethics and legislation, the absence of conflicts of interest, and transparency, which biobanks need to respect, notably thanks to the new international ISO 20387 norm for biobanking activity [21].

We established and developed a new collection dedicated to lymphoid malignancies within the Côte d’Azur Biobank (BB-0033-00025, University Côte d’Azur, Nice, France) to meet the needs of the different stakeholders undertaking research programs into lymphoid malignancies. Here we describe the strategy for setting up a lymphoma biobank. We give details of the samples included in this new collection, the indicators used, and the associated risks and factors to maintain sustainability. Finally, we propose a number of perspectives and directions for the use of this new collection for different research purposes.

## 2. Materials and Methods

### 2.1. Ethics, Consent, Informative Procedures, and Confidentiality

Between 1 November 2004, and 31 December 2022, we retrospectively selected all the patients diagnosed with a lymphoid malignancy within the laboratory of clinical and experimental pathology (LPCE, Nice University Hospital, Nice, France) for inclusion into the Côte d’Azur Biobank (BB-0033-00025) (www.biobank-cotedazur.fr (accessed on 14 February 2018)). The biobank was certified according to the ISO 9001 and ISO 20387 norms in 2022 and has been certified according to the S96-900 norm since 2011. The different biological resources collected in the biobank came from the LPCE, which has been accredited according to the ISO 15189 norm since 2011.

Patients samples were provided from different departments of the Nice University Hospital (Thoracic Oncology, Head and Neck, Ophthalmology and Dermatology). An informed consent form was signed both by the patient and the physician, either before sampling or retrospectively by e-mail or postal mail [22]. All data were anonymized and de-identified in our dataset, but the medical team is able to identify the patient in order to update information associated with each sample, notably the patient follow-up. When a patient was deceased at the time of inclusion, the samples could be included in the biobank without consent, except in cases of refusal of consent obtained during the patient’s lifetime or by his/her relatives.

All the tissue specimens were sampled in the LPCE and stored in the biobank among other pre-existing collections (thoracic, dermatology, thyroid, and head and neck collections) or stored for diagnostic purposes in the clinical pathology unit.

All procedures were conducted according to the Helsinki Declaration and have been approved by the local ethics committee (Human Research Ethics Committee, Centre Hospitalier Universitaire de Nice/Biobank BB-0033-00025/#DC-2022-5178, Nice, France).

### 2.2. Frozen Tissue Management

After resection, each tissue specimen was placed in a plastic sterile container (Dutscher, Brumath, France) and then in an envelope for vacuum transport (Decomatic, La Verpilliere, France) to the LPCE through a pneumatic tube. When possible, tumor fragments selected by a pathologist were frozen in liquid nitrogen (LN2) and limited to a total of 12 fragments for each tumor. This was done under a hood, with sterile gloves and razor blades, on a dedicated surface area cleaned with an RNAse-up solution bAU3 (Molecular BioProducts, San Diego, CA, USA). All scissors and forceps were cleaned with RNAse-up solution before use. All specimens were stored in cryotubes (1 specimen per tube) (1.8 mL, Nunc cryotube; Dutscher) and weighed immediately before freezing.

### 2.3. FFPE Tissue SPECIMEN Management

After resection, each tissue specimen was fixed in formalin solution, neutral buffered at 10%, for around 24 h prior to machine processing and paraffin embedding to obtain FFPE blocks using Tissue-Tek TEC 5 and Tissue-Tek VIP 6 (Sakura, Fujiyoshida City, Japan) devices according to the manufacturer’s instructions. The FFPE blocks for care were stored at room temperature. FFPE blocks dedicated to the biobank were stored at 4 °C.

### 2.4. Blood Sample Management

Blood samples were sent via pneumatic transport to the laboratory. Blood was collected in EDTA tubes (Vacuette^®^ EDTA K2, Greiner-Bio-One, Kremsmünster, Austria) when samples could be processed <60 min after sampling or in Cell-Free DNA BCT CE-IVD tubes (Streck, La Vista, NE, USA) in cases when the samples could not be processed within this short time. Centrifugation, aliquoting, processing, and freezing at −80 °C of samples of blood products was limited to a total of 12 tubes per patient and performed at the LPCE, as previously described [23,24]

### 2.5. Sample Management Workflow and Methodology for Inclusion into the Lymphoma Collection

All the tissue and biofluid specimens were obtained from one of the departments of the Nice University Hospital and then transferred to the LPCE to be assigned an ID number.

Extra frozen tissue sections, FFPE blocks, and tubes of blood products from patients diagnosed with lymphoma previously stored for care were retrospectively “reclassified” from care to research purposes after consent from the patients and included in the “lymphoma” collection within the biobank (#DC-2022-5178). Any extra lymphoma material stored in pre-existing collections was retrospectively transferred into the lymphoma collection. At least one FFPE block was always dedicated to care. Initially, samples are collected at the time of diagnosis when there is a strong clinical suspicion of lymphoma. Thus, most cases of reactive lymphoid hyperplasia were excluded from the biobanking process. However, if samples of lymphoid hyperplasia were inadvertently stored, we retained the material in the biobank, but we did not complete the dataset and did not report such cases in the manuscript (3 cases identified so far, Figure 1). When a diagnosis of lymphoma was made solely based on histology without clinical suspicion, we could retrospectively include formalin-fixed paraffin-embedded (FFPE) material in the biobank, but we lacked frozen tissue or blood samples. The flow of sample management is shown in Appendix A.

Any material included in the collection was assessed for a histological diagnosis, the biohazard status (HIV, HBV, and HCV) if available from the hospital clinical database, the percentage of tumor cells, the number and weight/volume of cryotubes, and the number of FFPE blocks included. The cold ischemia time was assessed for frozen tissue (time from resection to freezing). Matched normal tissue was not included in the biobank given that lymphoma lesions are diffuse and cannot be grossly distinguished from healthy tissue.

Histological validation by the “LYMPHOPATH” network was recorded when available. Briefly, in 2010, the French National Cancer Institute (INCa) created the LYMPHOPATH network to encourage pathologists in France to send every newly diagnosed or suspected lymphoma sample to a reference center for a second reading by an expert hematopathologist (https://www.e-cancer.fr (accessed on 24 May 2005). Both the final and submitted diagnoses are permanently recorded in the LYMPHOPATH database [25].

### 2.6. Clinico-Epidemiological Data and Patient Follow-Up

The following clinical, epidemiological, and biological data were recorded: age, sex, date of diagnosis, date and status at last follow-up, stage, diagnosis, location of disease, and level of LDH (the recorded items are available on demand). All clinical data are annually updated.

### 2.7. Statistics and Graphics

The Kaplan-Meier method with the log-rank test was used for the analysis of survival. A *p*-value of <0.05 was considered significant. Statistical tests and data representation were performed using GraphPad and STATAID software [26].

## 3. Results

### 3.1. Biological Resources

2150 samples of 363 cases (351 patients), including 1694 samples of 290 cases obtained at diagnosis, were listed, collected, and included in the lymphoma biobank (Figure 1 and Table 1). The collection was composed of 917 FFPE tissue blocks, 1010 frozen tissue cryotubes, 156 plasma cryotubes, and 67 other biological resources (Table 1). A total of 256 cases matched FFPE and frozen tissues, while 49 cases matched FFPE tissue and plasma samples. No biological resources were included without matched FFPE tissue. The median number of FFPE blocks included per case was 3 (ranging from 1 to 7) with 177 (49%) cases having ≥3 FFPE blocks (Table 1).

### 3.2. Histological Data

A total of 244 of 363 (67%) cases of B-cell lymphoid proliferations and lymphomas were included in the collection. Diffuse large B-cell lymphoma was the most frequent diagnosis (17%), followed by extranodal marginal zone lymphoma of MALT (39 cases, 11%). Of the 363 cases, 93 were (26%) classic Hodgkin lymphomas, along with 54 (15%) of the nodular sclerotic subtype. A total of 20 of 363 cases (6%) of T-cell and NK-cell lymphoid proliferations and lymphomas T were included, with T-lymphoblastic lymphoma (six cases) being the most frequent diagnosis, followed by peripheral T-cell lymphoma, NOS, extranodal NK/T-cell lymphoma, nasal-type, and nodal T-follicular helper (TFH) cell lymphoma, NOS (four cases for each diagnosis, respectively). The main locations of lymphoma were the mediastinum (36%), lymph node (23%), Waldeyer’s ring (14%), and lung (13%). Five cases (5) were diagnosed with lymphoid neoplasms, and 33% of cases were reread within the French LYMPHOPATH network. The description of all the lymphomas included in the collection is shown in Table 2 and Figure 2.

### 3.3. Clinical Data

Clinically, the median follow-up was 33 months [IQR:12–87]. The male/female ratio was 1.3. The median age at diagnosis was 58 years old [IQR:43–71]. A total of 104 of 363 (30%) patients were dead at the time of inclusion into the collection. Staging data were available for 77 patients and showed 43 (56%) advanced stages (stages III/IV). All the clinical characteristics of the patient are shown in Table 3. An univariate survival analysis showed the age at diagnosis, male gender, advanced stage, and lymphoma diagnosis were associated with shorter survival (Figure 3 and Appendix A).

## 4. Discussion

In this study, we established retrospectively a new lymphoma collection within the biobank of the University Côte d’Azur (Nice, France) and present the main criteria followed since its creation. The indicators used for lymphoma sample monitoring were defined by expert working groups [27,28], and have been simplified in this study. Over a period of 17 years, we systematically collected and registered information based on the following data:-Quantitative data: The number of patients who signed informed consent for the collection of their biological samples and clinically associated data. The number of FFPE blocks and the number and weight/volume of frozen tissues in cryotubes.-Qualitative data: Cold ischemia time before the tissue was frozen and processed. Delay between blood sampling, centrifugation, and storage at −80 °C in the laboratory. A block of FFPE tissue, taken as a mirror sample of the frozen sample, was cut and stained to confirm and quantify the tumor content of the corresponding frozen sample. These qualitative data reflected the handling of the pre-analytical phase [29,30,31].

Preservation is a subject of concern in a biobank and needs to be monitored and improved continuously. All FFPE samples transferred from pre-existing collections to the lymphoma collection were stored at 4 °C and protected from light and humidity to ensure better conservation of protein epitopes and nucleic acids (especially RNA) for further analysis [32,33].

To our knowledge, we have developed here the first neoplasia collection specifically dedicated to academic and industrial research, including blood, frozen tissue, and FFPE material from different types of Hodgkin’s and non-Hodgkin’s lymphomas. Although some lymphoma collections exist or are being set up in Western countries (e.g., the French TENOMIC collection for T-cell lymphoma, the French GHEDI collection for diffuse large B-cell lymphoma, and British and American mantle cell lymphoma cell banks), they are often associated with clinical programs and dedicated to academic research [34,35,36,37,38,39]. In addition, collections of lymphomas specifically dedicated to establishing partnerships with industry in France (e.g., CeVi and CeVi-CAR T collections—https://experts-recherche-lymphome.org/calym/explorer-les-ressources-du-consortium/collection-cevi/ (accessed on 8 December 2021)) are comprised only of samples of viable cells. In Europe, the Swedish U-CAN national biobank is a large cancer biobank, including cancer tissue and blood samples, notably a lymphoma collection [40]. However, projects are required to include at least one local representative from the U-CAN diagnosis group. Regarding Asian biobanks, the BioBank Japan Project and the China Kadoori biobank are huge governmental biobanks with very heterogeneous biological material dedicated to genome-wide association studies [41,42].

Our collection has specific epidemiological features, given that our samples preferentially come from thoracic and head and neck oncology units. The collection is enriched in mediastinal lymphomas (Primary Mediastinal B-cell Lymphoma and Hodgkin lymphoma) [43,44], with a younger age at diagnosis of the cohort (median age at diagnosis of 57 years with less than half of the patients being >60 years old) compared to an average age at diagnosis of 67 years old with 55–60% of patients being >65 years old [45,46]. Accordingly, the collection is also enriched in lymphomas with preferential lung (MALT lymphoma) and head and neck locations (extra-osseous plasmocytoma) [47,48]. Moreover, follow-up data seems to be consistent with known prognostic factors previously described for lymphoma, except for the LDH level, which was difficult to collect as different techniques and thresholds have been used in our institution over time [49,50,51]. However, the survival data analysis should be interpreted with care due to the heterogeneity of lymphoma subtypes included in the biobank and the relatively long period of inclusion. Thanks to this, we recently initiated a research program together with Thermo-Fischer Scientific to develop NGS panels dedicated to the management of lymphoma (Liverpool Pan Lymphoid NGS Panel, contact: geetha.menon@liverpoolft.nhs.uk; lihui.wang@liverpoolft.nhs.uk). We were able to run specific mediastinal lymphoma samples for technical validation of the NGS panel. This project will help characterize lymphoid malignancies at a genomic level, which is crucial to understanding lymphomagenesis and developing innovative diagnostic tools and therapies [52,53]. In the literature, a few studies showed that the analysis of biobank samples has helped research by facilitating tumor stratification and revealing novel biomarkers in lymphoma [54,55]. Along this line, our collection aims to integrate local and international academic research programs and facilitate partnerships with industry in the field of lymphoma for better management of this rare disease.

Another specific aspect of the diagnosis of lymphoma in France is the LYMPHOPATH network. This network significantly contributes to a precise diagnosis of lymphoma and optimal clinical management in a proportion of patients [25]. It is worth noting that the LYMPHOPATH network has validated only 33% of our cases. However, a part of the inclusion period for the collection started before the development of this network (2010), which may explain this result.

Globally, several projects associated with the discovery and validation of several biomarkers have emerged since the establishment in 2006 of our biobanks, mainly associated with our thoracic collection [56]. The involvement in the past of six senior pathologists in the functioning of the biobank has been beneficial in contributing to projects into lung cancer, the largest collection of our biobank [57]. The increase in the number of projects and in the release of samples associated with a material transfer agreement have allowed sustainability and optimization of the biobank, thereby enabling the development of the “lymphoma” collection. Moreover, certification of the biobank located in the pathology laboratory of the hospital and accredited according to ISO 9001, 15189, and, especially, 20387 norms are important factors for its recognition by academic and industrial partners.

However, several issues regarding the implementation of this new collection need to be pointed out.

The retrospective collection by e-mail or postal mail of consent from patients is the first limitation, as many are still pending, making their status unknown and hindering inclusion into the biobank. The mistrust of patients towards the use of data may also further delay research programs. Therefore, disseminating widely the crucial impact of biobanks on cancer research to the medical community and patients can encourage systematic consent during tumor sampling.

In addition, retrospective data collection is time-consuming and results in the loss of crucial information linked to the samples. Thus, starting a biobank collection as soon as possible is important to begin the prospective inclusion process. The current absence and limited availability of certain biological sources of interest (feces, cerebrospinal fluid, saliva, and urine) could weaken the aims of the collection.

Finally, the need to collect in a prospective way increasing amounts of complex data (e.g., genome sequencing data) and the difficulties of connecting complex information obtained from novel morphological analyses (chromogenic multiplexing) remain a huge challenge [58].

Maintaining a biobank’s sustainability is a significant challenge, as it requires the preservation of existing collections such as the Lymphoma collection and the provision of high-quality annotated biological resources on request. In addition to implementing control measures in management through achievable strategic orientation and performance indicators [27] we also need to ensure the financial and sustainable viability of the biobank [59,60,61]. The Nice University Hospital Biobank has dedicated a considerable amount of expense towards human and material resources, achieving an annual average of nearly 60% of the annual revenue through publications, public and private partnerships, and the transfer of biological resources [56]. Despite being a long process, biobanking of collections remains the only way to provide quality material and clinically associated data for rigorous research protocols and should be developed in each institution. Prioritizing educational programs like the “Biobanks and Complex Data Management” Master of Sciences (University Côte d’Azur, Nice) can raise awareness among the younger generation (https://univ-cotedazur.eu/msc/biobanks-complex-data-management#.Xmj1ukGDNp8 (accessed on 29 October 2020)).

The future plans for the BB-003-00025 biobank involve the incorporation of more intricate data related to the biological samples. The biobank’s database should be capable of recording diverse information obtained from future research projects that use these samples, such as whole genome sequencing. Another aspect that can be explored, thanks to the prospective inclusion of samples, is the collection of sequential biopsies during treatment monitoring, which could enhance the development of projects that seek new prognostic and predictive biomarkers. In addition, this approach presents an opportunity to expand the scope of sample collection with non-lymphoma samples (reactive lymphoid hyperplasia). Finally, digitizing pathology slides and making them available to stakeholders could facilitate the growth of biobank research programs, making it an interesting avenue to pursue [62].

## 5. Conclusions

In conclusion, establishing a new collection of lymphoma samples depends on many actors and supplies to ensure good quality material stored in appropriate conditions, as well as on obtaining clinically associated data, including patient follow-up. Setting up such a collection requires time, team expertise, and knowledge to meet the needs of the stakeholders. This collection is expected to be prospectively updated and enlarged in the future through the addition of new samples of lymphomas as well as reactive lymphoid tissue. The purpose of this original collection was to create a resource that could be used by and disseminated among researchers to support high-quality research programs, contributing to improved comprehension and management of lymphoma.

## Figures and Tables

**Figure 1 jpm-13-01076-f001:**
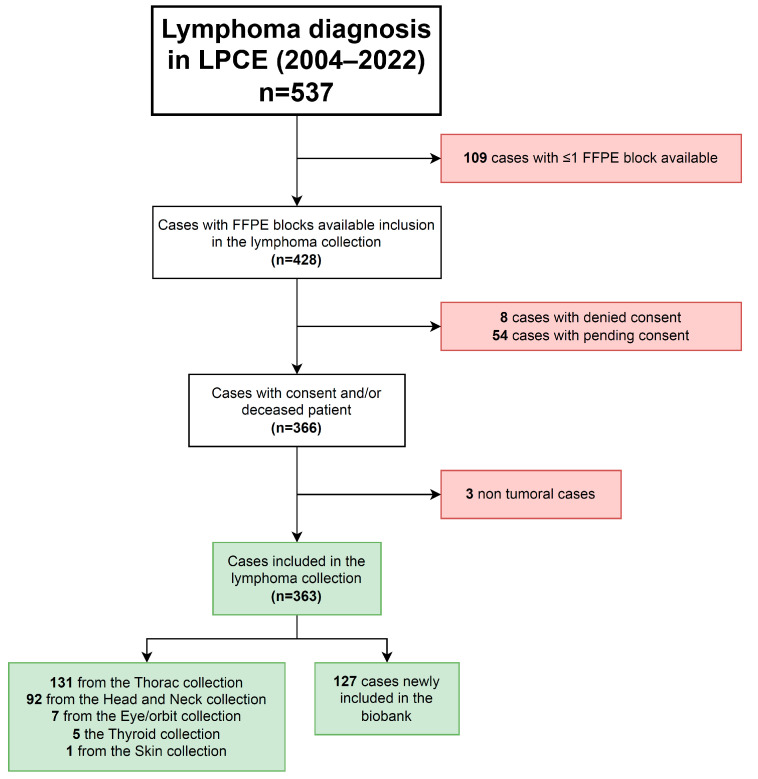
Flow chart of the cohort of cases included in the lymphoma collection of the biobank.

**Figure 2 jpm-13-01076-f002:**
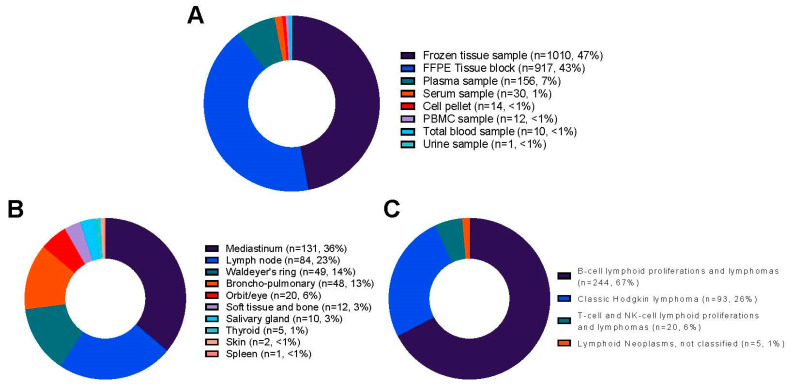
Distribution of the 2150 samples from 363 cases included in the lymphoma collection of the biobank: (**A**) Plot depicting the type of samples; (**B**) Plot depicting the lymphoma cases; (**C**) Plot depicting the main classification of the lymphoma cases. NOS: Not otherwise specified.

**Figure 3 jpm-13-01076-f003:**
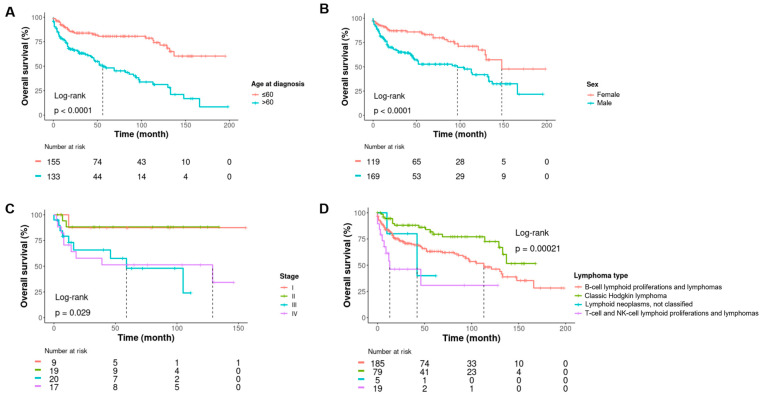
Kaplan-Meier curves showing estimates of overall survival with a univariate log-rank analysis for patients according to age at diagnosis (**A**), sex (**B**), stage (**C**), and type of lymphoma (**D**).

**Table 1 jpm-13-01076-t001:** Biological resources included in the lymphoma collection.

	All Samples (n = 2150)
FFPE blocks	
Median per cases [IQR]	3 [2–3]
At diagnosis	741
Total	917
Frozen tissue samples	
Median per cases [IQR]	4 [2–5]
At diagnosis	810
Total	1010
Plasma samples	
Median per cases [IQR]	2 [2–4]
At diagnosis	104
Total	156
Serum samples	
Median per cases [IQR]	2 [2–2]
At diagnosis	18
Total	30
Cell pellet	
Median per cases [IQR]	2 [2–2]
At diagnosis	8
Total	14
PBMC samples	
Median per cases [IQR]	2 [2–2]
At diagnosis	8
Total	12
Total blood samples	
Median per cases [IQR]	1 [1–2]
At diagnosis	5
Total	10

IQR: inter-quartile range.

**Table 2 jpm-13-01076-t002:** Histological and biological characteristics of the cases included in the biobank.

	All Cases (n = 363)
Consent	
Unknown	70 (19)
Yes	293 (81)
Send by e-mail or postal mail, waiting a reply	54
Cases included at diagnosis	290 (80)
Cases included at 1st relapse	69 (19)
Cases included at multiple relapse	4 (1)
FFPE Blocks per case, median [IQR]	3 [2–3]
Cases with ≥3 FFPE blocks	177 (49)
% of tumor cells, median [range] (*n* = 240)	70 [1–100]
Cold ischemia, median [IQR], minute (*n* = 228)	20 [15–25]
≤15	102 (45)
>15 and ≤30	87 (38)
>30 and ≤45	33 (14)
>45	6 (3)
Matched frozen tissue	256 (71)
Matched plasma	49 (14)
Matched serum	16 (4)
Matched PBMC	6 (2)
Matched total blood sample	7 (2)
Matched cell pellet	7 (2)
LYMPHOPATH network rereading	120 (33)
Tumor location	
Mediastinum	131 (36)
Lymph node	84 (23)
Waldeyer’s ring	49 (14)
Broncho-pulmonary	48 (13)
Orbit/eye	20 (6)
Soft tissue and bone	12 (3)
Salivary gland	10 (3)
Thyroid	5 (1)
Skin	3 (0.8)
Spleen	1 (0.3)
Diagnosis	
B-cell lymphoid proliferations and lymphomas	244 (67)
Precursor B-cell neoplasm	1 (0.3)
B-lymphoblastic leukemia/lymphoma, NOS	1 (0.3)
Mature B-cell neoplasms	243 (67)
Diffuse large B-cell lymphoma, NOS	63 (17)
Extranodal marginal zone lymphoma of MALT	39 (11)
Indolent B-cell lymphoma, not classified	30 (8)
Primary mediastinal B-cell lymphoma	23 (6)
Follicular lymphoma	23 (6)
Extraosseous plasmacytoma	14 (4)
Chronic lymphocytic leukemia/small lymphocytic lymphoma	14 (4)
Mantle cell lymphoma	11 (3)
Nodular lymphocyte predominant B-cell lymphoma	7 (2)
Large B-cell lymphoma, not classified	4 (1)
Nodal marginal zone lymphoma	4 (1)
Mediastinal Grey Zone Lymphoma	3 (0.8)
Plasmablastic lymphoma	2 (0.6)
Mature B-cell neoplasm, not classified	2 (0.6)
Lymphoplasmacytic lymphoma	1 (0.3)
Lymphomatoid granulomatosis	1 (0.3)
Burkitt Lymphoma	1 (0.3)
Diffuse large B-cell lymphoma associated with chronic inflammation	1 (0.3)
Classic Hodgkin lymphoma	93 (26)
Nodular sclerosis	54 (15)
Not classified	31 (9)
Mixed cellularity	4 (1)
Lymphocytes rich	3 (0.8)
Lymphocytes depleted	1 (0.3)
T-cell and NK-cell lymphoid proliferations and lymphomas	20 (6)
Precursor T-cell neoplasms	
T-lymphoblastic leukemia/lymphoma, NOS	6 (2)
Mature T-cell and NK-cell neoplasms	14 (4)
Peripheral T-cell lymphoma, NOS	4 (1)
Extranodal NK/T-cell lymphoma, nasal-type	4 (1)
ALK-negative anaplastic large cell lymphoma	4 (1)
Nodal T-follicular helper (TFH) cell lymphoma, NOS	2 (0.6)
Lymphoid neoplasms, not classified	5 (1)

FFPE: formalin-fixed paraffin embedded; IQR: inter-quartile range; PBMC: peripheral blood mononuclear cells; NOS: not otherwise specified.

**Table 3 jpm-13-01076-t003:** Clinical and follow-up characteristics of the patients included in the lymphoma collection.

	All Patient (n = 351)
Patient with full dataset	91 (25)
Follow-up, median [IQR], m	33 [12–87]
Age at diagnosis, median [IQR], y	58 [43–71]
>60	166 (47)
LDH count at diagnosis (U/i), median [IQR]	446 [352–581]
Elevated LDH at diagnosis	55 (43)
Status at last follow-up	
Alive	247 (70)
Dead	104 (30)
Sex	
Female	152 (43)
Male	199 (57)
Stage at diagnosis	
I	13 (17)
II	21 (27)
III	23 (30)
IV	20 (26)

m: month; y: year; IQR: inter-quartile range.

## Data Availability

The data presented in this study are available on request from the corresponding author.

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
