# Peer review of "The Need to Set up a Biobank Dedicated to Lymphoid Malignancies: Experience of a Single Center (Laboratory of Clinical and Experimental Pathology, University Côte d’Azur, Nice, France)"

_jpm, 2023, doi:10.3390/jpm13071076_

Round 1
Reviewer 1 Report
Bontoux and colleagues report on a single center biobank for malignant lymphomas including FFPE, fresh frozen and different types of blood samples. This topic is highly important, as clinically annotated biobanks are more and more important e.g. for evaluation of prognostic, predictive, and theranostic biomarkers. However, the reviewer has some points to be addressed:
Major:
1) The report should include the time points when the biomaterials were obtained, e.g. at diagnosis, first relapse etc. It should be given how many fresh frozen, FFPE etc. materials were obtained at diagnosis. What were the time points of blood samples? What percentage of patients had a full dataset to a given time point?
2) The authors described that the biobank is academic and industry sponsored. What interests are depicted in the consent form? Does the biobank sell materials and data (anonymized/ pseudonymized) to partners? Are analyses outside of the EU possible?
3) The authors described on the one hand side that the data were anonymized but on the other hand side annual updates of clinical data were made? Please clarify.
4) Has the biobank also a quality standard or SOP regarding freeze/ thaw cycles of biosamples? What system is used to track the samples? Does the biobank track samples if sent for project to partners?
5) How did the biobank handle non-lymphoma-samples (eg. at diagnosis)? Were these samples excluded from the biobank.
Minor errors should be corrected.
Reviewer 2 Report
This study reports the establishment of a lymphoma biobank which contains 2150 samples from 351 patients (tissue specimens and matched biological resources) and associated clinical data. They describe the strategy of developing the lymphoma biobank, the details of the samples, the indicators for use, the associated risk and factors to maintain sustainability, and also discuss the perspectives and directions for future use of this collection.
Comments:
1.The biobank is not well representative of all the lymphoma types. The classifications for some lymphoma types are not specific or do not follow the WHO and ICC classifications. For example, “Gray zone lymphoma”- need to be specified? “Aggressive B-cell non-Hodgkin lymphoma”. “Sclero-nodular CHL”- Nodular sclerosis type?
2.About 20% of the cases are not further classified, including “Low grade B-cell non-Hodgkin lymphoma, NOS”, “B-cell non-Hodgkin lymphoma, NOS”, “CHL, NOS”, “Lymphoma NOS”. What’s the difference between “Low-grade B-cell non-Hodgkin lymphoma, NOS” and “B-cell non-Hodgkin lymphoma, NOS”. I am not aware of “CHL, NOS” being a subtype of CHL. Is “Lymphoma, NOS” a gray zone lymphoma or composite lymphoma or others?
3.The lymphomas and their subtypes are heterogeneous in terms of prognosis and overall survival. I don’t think it’s a good idea to study their overall survival as a whole.
4.Please add the number of cases for “Lymphoma, NOS” in table 2.
Round 2
Reviewer 2 Report
No additional comments.